# Genome-Wide Identification and Analysis of the *MADS-Box* Gene Family in Almond Reveal Its Expression Features in Different Flowering Periods

**DOI:** 10.3390/genes13101764

**Published:** 2022-09-29

**Authors:** Xingyue Liu, Dongdong Zhang, Zhenfan Yu, Bin Zeng, Shaobo Hu, Wenwen Gao, Xintong Ma, Yawen He, Huanxue Qin

**Affiliations:** College of Horticulture, Xinjiang Agriculture University, Urumqi 830052, China

**Keywords:** *Prunus dulcis*, *MADS-box* gene family, evolutionary analyses, expression patterns, flowering

## Abstract

The *MADS-box* gene family is an important family of transcription factors involved in multiple processes, such as plant growth and development, stress, and in particular, flowering time and floral organ development. Almonds are the best-selling nuts in the international fruit trade, accounting for more than 50% of the world’s dried fruit trade, and one of the main economic fruit trees in Kashgar, Xinjiang. In addition, almonds contain a variety of nutrients, such as protein and dietary fiber, which can supplement nutrients for people. They also have the functions of nourishing the yin and kidneys, improving eyesight, and strengthening the brain, and they can be applied to various diseases. However, there is no report on the *MADS-box* gene family in almond (*Prunus dulcis*). In this study, a total of 67 *PdMADS* genes distributed across 8 chromosomes were identified from the genome of almond ‘Wanfeng’. The *PdMADS* members were divided into five subgroups—Mα, Mβ, Mγ, Mδ, and MIKC—and the members in each subgroup had conserved motif types and exon and intron numbers. The number of exons of *PdMADS* members ranged from 1 to 20, and the number of introns ranged from 0 to 19. The number of exons and introns of different subfamily members varied greatly. The results of gene duplication analysis showed that the *PdMADS* members had 16 pairs of segmental duplications and 9 pairs of tandem duplications, so we further explored the relationship between the *MADS-box* gene members in almond and those in *Arabidopsis thaliana*, *Oryza sativa*, *Malus domestica*, and *Prunus persica* based on colinear genes and evolutionary selection pressure. The results of the *cis*-acting elements showed that the *PdMADS* members were extensively involved in a variety of processes, such as almond growth and development, hormone regulation, and stress response. In addition, the expression patterns of *PdMADS* members across six floral transcriptome samples from two almond cultivars, ‘Wanfeng’ and ‘Nonpareil’, had significant expression differences. Subsequently, the fluorescence quantitative expression levels of the 15 *PdMADS* genes were highly similar to the transcriptome expression patterns, and the gene expression levels increased in the samples at different flowering stages, indicating that the two almond cultivars expressed different *PdMADS* genes during the flowering process. It is worth noting that the difference in flowering time between ‘Wanfeng’ and ‘Nonpareil’ may be caused by the different expression activities of *PdMADS47* and *PdMADS16* during the dormancy period, resulting in different processes of vernalization. We identified a total of 13,515 target genes in the genome based on the MIKC DNA-binding sites. The GO and KEGG enrichment results showed that these target genes play important roles in protein function and multiple pathways. In summary, we conducted bioinformatics and expression pattern studies on the *PdMADS* gene family and investigated six flowering samples from two almond cultivars, the early-flowering ‘Wanfeng’ and late-flowering ‘Nonpareil’, for quantitative expression level identification. These findings lay a foundation for future in-depth studies on the mechanism of *PdMADS* gene regulation during flowering in different almond cultivars.

## 1. Introduction

Plant growth and development are mainly divided into two processes: vegetative growth and reproductive growth [1]. After a certain period of vegetative growth, a series of reproductive growth processes occur, producing flower buds, flowers, fruits, and seeds [2]. For flowering plants, flowering is an important sign of the beginning of the reproductive growth process, signifies the end of vegetative growth, and is also one of the basic requirements for plant reproduction [3]. Flowering is affected by both internal and external factors. Various factors, such as the external temperature, humidity, photoperiod, and circadian rhythm changes, as well as internal photosynthetic product accumulation, mineral element absorption, and changes in hormone levels, affect the flowering time [4]. Internal and external factors coordinate and interact with each other to influence the flowering process. Therefore, in response to the effects of these factors on flowering, plants have evolved various mechanisms, such as vernalization and circadian rhythm, to regulate the flowering process [5]. Additionally, plants alter their flowering times in response to adverse environmental impacts [6]. Moreover, flowering is an important trait in plant breeding and engineering [7]. Overall, in-depth study of the genes that regulate flowering is of great significance for plant breeding and agricultural production [8].

In plants, genome-wide identification can identify and classify genes, facilitate the discovery of new genes, and also provide a better understanding of their molecular evolution. As more and more plants conduct genome-wide identification of *MADS-box*, the response mechanism of the *MADS-box* gene in plant growth and development, flowering, and response to stress has been continuously researched [9]. The *MADS-box* gene family is one of the most widely studied families in fungi, plants, and animals [9]. ‘MADS’ is named after the *minichromosome maintenance* gene (*MCMI*) of yeast, the *AGAMOUS* (*AG*) gene of *A. thaliana*, the *DEFICIENCE* (*DEF*) gene of snapdragon, and the human serum response factor (*SRF*) [10]. *MADS-box* gene sequences have a conserved N-terminal DNA binding domain of approximately 60 amino acids, which binds to the CArG box and is widely involved in regulating eukaryotic growth, development, and signal transduction [11,12]. To date, among type-I genes, only 4 Mα subfamily (*AGL23*, *AGL28*, *AGL61*, and *AGL62*) and 2 Mγ subfamily (*AGL37* and *AGL80*) genes, which play important roles in seed development, have been identified in *A. thaliana* [13]. Type-II genes, which include those of the Mδ and MIKC subfamilies, are classified according to their structural characteristics, and their protein domains include four types: MADS, K, I, and C terminal [14]. The MIKC^C^ type includes 14 subclasses: AP1, AP3, PISTILLATA (PI), AGAMOUS (AG) or SEEDSTICK (STK), AGAMOUS-LIKE6 (AGL6), AGL12, AGL15, AGL17, BSISTER (BS), SUPPRESSOR OF OVEREXPRESSION OF CONSTANS1 (SOC1), SHORT VEGETATIVE PHASE (SVP), FLOWERING LOCUS C (FLC), and SEPALLATA1 (SEP1) [15].

The almond (*P. dulcis*) is one of the most important dried fruits in the world and one of the most important economic fruit trees in Xinjiang, China [16]. Almonds are native to the mountainous areas of West and Central Asia, and planting areas are widely distributed [17]. Therefore, almond trees in different cultivation areas have significant external and internal differences. In our previous study, we found a significant difference in flowering time between Xinjiang’s native cultivar ‘Wanfeng’ and the American cultivar ‘Nonpareil’. Indeed, ‘Wanfeng’ reached the full bloom period 25–30 days earlier than ‘Nonpareil’. According to the statistical results of the time to reach full bloom across 6 years (2016–2021), 50% of all flowers on a ‘Wanfeng’ tree reached a state of full flowering between stages 3.20 and 3.26 each year, while those for ‘Nonpareil’ reached full flowering between stages 4.23 and 4.30 each year. *MADS-box* family genes play an important role in regulating the flowering process. Therefore, this study used the genome-wide data of ‘Wanfeng’ almond obtained by our team to identify *MADS-box* genes. Bioinformatics methods were used for protein physicochemical property analysis, phylogenetic tree construction and classification, the conserved motif, gene structure, gene location, gene duplication, and *cis*-acting element identification and expression pattern analysis of almond MADS family members at different flowering stages. Related genes were selected from the MIKC subfamily, and fluorescence quantitative expression analysis across six flowering stages was carried out in the two cultivars of ‘Wanfeng’ and ‘Nonpareil’. The results of this study will help our understanding of the evolution and function of almond *MADS-box* genes, especially regarding preliminary exploration of the characteristics of these genes in regulating almond flowering, and provide a theoretical basis for subsequent research on the regulation of flowering in almonds.

## 2. Materials and Methods

### 2.1. Identification of MADS-Box Family Members in the Whole Genome of ‘Wanfeng’ Almond

The *MADS-box* genes were identified using the genome-wide data of ‘Wanfeng’ almond obtained by our team (BioProject ID: PRJNA854307). First, we downloaded the SFR (PF00319) and MEF2 (PF09047) hidden Markov models through the Pfam database (http://pfam.xfam.org/ (accessed on 17 July 2022)) [18]. The HMMER tool was used to search and align the whole-genome protein sequence of ‘Wanfeng’ almond with the two hidden Markov models, and protein sequences with an E-value ≤ e−5 were retained [19]. The whole-genome protein sequences were then aligned with *Arabidopsis MADS-box* protein sequences by the BLASTP tool, and protein sequences with an E-value ≤ e−5 were retained. The HMMER and BLASTP screening results were combined to remove redundant and repetitive sequences, and domain validation was performed on the Pfam data. Finally, 67 *MADS-box* genes in the ‘Wanfeng’ almond genome were identified. Prediction of the protein physicochemical properties was performed by ExPASy (http://web.expasy.org/protparam/ (accessed on 17 July 2022)) and WoLF PSORT II (https://www.genscript.com/wolf-psort.html?src=leftbar (accessed on 17 July 2022)) subcellular localization analysis [20,21].

### 2.2. Phylogenetic Tree Construction and Classification

The protein sequences of 104 *A. thaliana MADS-box* genes were downloaded from the UniProt database (https://www.uniprot.org/ (accessed on 18 July 2022)). We performed MUSCLE multiple alignment of *Arabidopsis* and almond *MADS-box* family member protein sequences using the MEGA X tool and phylogenetic tree construction using neighbor-joining with 1000 bootstrap replications with the Poisson correction model and pairwise deletion [22]. An evolutionary tree was built, and subfamilies were classified using the Evolview tool.

### 2.3. Motif, Domain, and Gene Structure Composition Analyses

The protein sequence motifs of the almond *MADS-box* family members were searched by the MEME (http://meme-suite.org/tools/meme (accessed on 20 July 2022)) online tool [23,24]. The number of motifs was set to 10, and the default values were used for other parameters. The location and number of MADS domains in the protein sequences of almond *MADS-box* family genes were analyzed by the NCBI CDD (https://www.ncbi.nlm.nih.gov/cdd/ (accessed on 20 July 2022)) online tool. The exon and intron positions and number of members of the *MADS-box* family in almond were assessed using general feature format (GFF) data. Finally, the phylogenetic tree, motifs, domains, and gene structures of almond *MADS-box* family members were clustered and mapped using TBtools [25].

### 2.4. Chromosomal Location, Gene Duplication, and Ka/Ks Analyses

The chromosomal and positional information of the 67 *MADS-box* family members was obtained from the GFF data for ‘Wanfeng’ almond, and TBtools was used to map the distribution of the genes on the chromosomes. Segmental duplication and tandem duplication gene analyses were performed on the *MADS-box* members with the MCScanX tool, and the collinear circle diagram of the chromosome segmental duplication gene pairs was drawn using the Circos tool [26,27]. In addition, we identified the collinear *MADS-box* family genes in almond and four other species, namely *A. thaliana*, *O. sativa*, *M. domestica*, and *P. persica*. Finally, the Ka, Ks, and Ka/Ks values were calculated for segmental duplication, tandem duplication, and the collinear genes using the Ka/Ks Calculator tool [28].

### 2.5. Identification of Promoter cis-Acting Elements

The 2000-bp upstream sequences of the *MADS-box* family members in almond were extracted by TBtools, and the *cis*-acting elements in these promoter regions were analyzed with the PlantCARE online tool [29]. The distribution map of the *cis* elements of *MADS-box* genes was constructed by TBtools and modified and analyzed using Adobe Illustrator CC 2019.

### 2.6. Analysis of the Expression Patterns of ‘Wanfeng’ and ‘Nonpareil’ MADS-Box Family Members at Six Flowering Stages

The expression patterns of the *PdMADS* members were analyzed by using the corresponding six flowering stage transcriptomes of ‘Wanfeng’ and ‘Nonpareil’ cultivated almond obtained by our team (GSA number: CRA007615 and CRA007633). First, the cuffLinks tool was used to calculate the fragments per kilobase of exon model per million mapped fragments (FPKM) values of 31711 transcripts in 6 flowering stage transcriptomes of ‘Wanfeng’ and ‘Nonpareil’ cultivated almond, from which the FPKM values of 67 *PdMADS* genes in 12 samples of A1~A6 and B1~B6 were obtained. Secondly, the *PdMADS* gene whose FPKM value was less than 1 in the 12 flowering stage samples was removed. Finally, 36 *PdMADS* members were retained for expression pattern analysis.

### 2.7. Material Handling and qRT–PCR Analysis

To explore the expression levels of *PdMADS* genes at different flowering stages of almond, 6-year-old ‘Wanfeng’ and ‘Nonpareil’ almonds were sampled from the almond resource garden of Shache county from 15 March to 30 April 30 2022. Using three robust trees, annual fruiting branches facing the south and northwest directions were selected from the middle of the crown to collect materials at six flowering stages (Appendix A): the dormant stage (A1 and B1), sprouting stages (A2~A5 and B2~B5), and full bloom stage (A6 and B6). The six flowering stage samples of each tree were stored separately with three biological replicates at −80 °C for future use.

We selected 15 representative *PdMADS* genes from the MIKC subfamily for qRT–PCR analysis of ‘Wanfeng’ and ‘Nonpareil’ at six flowering stages. The qRT–PCR primers for the selected *PdMADS* genes were designed by Primer Premier 5 (Appendix A). The total RNA was extracted from frozen embryonic samples by using the RNAprep Pure Plant Kit (TIANGEN, Beijing, China). The reaction system (25 mL) consisted of the following: 10 mL of SYBR Green PCR Master Mix, 0.4 mL of forward primer (10 mmol/L), 0.4 mL of reverse primer (10 mmol/L), 5 mL of diluted cDNA (50 ng/mL), and 25 mL of RNase-free water [30]. The reaction conditions were as follows: predenaturation at 95 °C for 3 min, with 40 cycles of 95 °C for 10 s, 55 °C for 20 s, and 72 °C for 20 s, followed by a final step of 75 °C for 5 s. The plates were then read to detect fluorescence signals (40 cycles). Melting curve analysis was conducted in a temperature range of 65–95 °C with a temperature increment of 0.5 °C/5 s [31]. The obtained cycle threshold (CT) values were quantitatively analyzed by the 2^−ΔΔCT^ method (Appendix A) [32].

### 2.8. Protein Interaction Network Construction and Enrichment Annotation

Almond MADS-box proteins were uploaded to the STRING database (https://string-db.org/ (accessed on 22 July 2022)) for node comparison, and the relationship between the almond MADS-box members was predicted based on *A. thaliana* protein interactions (Appendix A). Gene Ontology (GO) and Kyoto Encyclopedia of Genes and Genomes (KEGG) enrichment annotations were performed on the interacting network proteins (Appendix A).

### 2.9. MIKC Target Gene Prediction and Enrichment Annotation

Based on the protein interaction prediction results, we downloaded the MIKC DNA-binding site (MA1203.1) from the c_CORE database (http://jaspar.genereg.net (accessed on 22 July 2022)) [33]. To obtain possible target genes regulated by MIKC, the 2000-bp upstream promoter sequences of the almond genes were extracted by TBtools. Then, the Motif FIMO (https://meme-suite.org/meme/ (accessed on 22 July 2022)) tool was used to detect the binding motifs in the almond promoters for MIKC. The final candidate target genes were determined based on the screening criteria of *p*-value < 1 × 10^−6^ [30]. The candidate MIKC target genes were subjected to GO and KEGG enrichment analysis using the TBtools tool. Finally, protein domain annotation of the candidate target genes was performed with the InterPro database (https://www.ebi.ac.uk/interpro/ (accessed on 25 July 2022)).

## 3. Results

### 3.1. Characteristic Analysis of PdMADS Family Members

A total of 67 *MADS-box* genes were identified in the whole genome of ‘Wanfeng’ almond, and each protein sequence contained at least one SRF domain. The 67 *MADS-box* genes were renamed *PdMADS1*~*PdMADS67* according to their chromosomal locations. The physicochemical properties of the *PdMADS* family members were also analyzed (Appendix A). The results showed that the protein sequence length ranged from 71 aa (PdMADS4) to 689 aa (PdMADS41), and the protein molecular weight ranged from 8.121 kDa (PdMADS4) to 78.059 kDa (PdMADS41). The isoelectric points ranged from 4.05 (PdMADS38) to 10.12 (PdMADS63), and 50 (pI > 7) and 17 (pI < 7) members of PdMADS were considered basic and acidic proteins, respectively. The PdMADS family member protein GRAVY (grand average of hydropathicity) was less than zero, indicating a hydrophilic protein. The results of the protein subcellular prediction of PdMADS family members showed that 52, 6, 5, and 4 members localized to the nucleus, chloroplast, cytoplasm, and mitochondrial matrices, respectively (Appendix A).

### 3.2. Phylogenetic Tree and Classification Analyses of PdMADS Family Members

To explore the clustering relationship and subfamily classification of *PdMADS* family members, we constructed a neighbor-joining phylogenetic tree of the protein sequences of the *MADS-box* members in almond and *A. thaliana*. The phylogenetic tree was divided according to the subgroup classification of *A. thaliana MADS-box* members and formed five subgroups: Mα, Mβ, Mγ, Mδ, and MIKC, with 17, 0, 17, 3, and 30 members, respectively (Figure 1A). In addition, we further divided the *PdMADS* members in the MIKC subfamily into 12 subgroups: SEP, FUL, FLC, SOC1, AGL11, AGAMOUS, APETALA3, PISTILLATA, ANR1, SVP, TT16, and AGL12, with 5, 3, 1, 3, 3, 2, 2, 1, 4, 4, 0, and 1 *PdMADS* genes, respectively (Figure 1B).

### 3.3. Motif, Domain, and Gene Structure Analyses of PdMADS Family Members

We further constructed a neighbor-joining phylogenetic tree of *PdMADS* family members (Figure 2A). A total of 10 motifs were annotated in the *PdMADS* family member protein sequences by the MEME online tool and clustered with the phylogenetic tree (Figure 2B). The results show that the protein sequences of *PdMADS* members in the same subfamily had highly similar motifs. The MIKC subfamily *PdMADS* members included motifs 1, 2, 4, 7, and 9, and all members had motif 1. The Mα and Mδ subfamily *PdMADS* members mainly showed conservation of motif 1 and motif 2. Members of the Mγ subfamily contained motifs 1, 3, 5, 6, and 8, and most of the members had motif 3. Among them, motifs 1, 3, and 5 were the SRF type, motif 2 was the MEF2 type, and motif 4 was the K-box type (Figure 2E). In addition, we further calculated the position and number of MADS domains of the *PdMADS* family members. PdMADS15 had two MADS domains, which was consistent with the motif results. All other protein sequences had only one MADS domain, which was primarily located at the N-terminus of the protein sequence (Figure 2C).

To explore the gene structure characteristics of the *PdMADS* family members, we assessed the number of exons and introns of all genes and clustered them within a phylogenetic tree (Figure 2D). The results show that the *PdMADS* genes of the same subfamily had similar numbers of exons and introns. Members of the Mα and Mγ subfamilies also had similar gene structures, except for *PdMADS61*, which had 3 exons and 2 introns, whereas the other genes had 1~2 exons and 0~1 introns. The *PdMADS* genes in the MIKC subfamily had 3~17 exons and 2~16 introns, and most members had 8 exons and 7 introns. Members of the Mδ subfamily had 10~20 exons and 9~19 introns.

### 3.4. Chromosomal Location, Gene Duplication, and Ka/Ks Analyses of PdMADS Family Members

The chromosomal locations of the *PdMADS* family members were determined based on the chromosomal gene distribution in the ‘Wanfeng’ almond genome. In addition, we set the genetic spacing to 200 kb to calculate the gene density per chromosome (Appendix A). This is represented by a color gradient from blue (low gene density) to red (high gene density) in Figure 3. Empty regions on the chromosomes indicate regions lacking gene distribution information. As shown in Figure 3, the 67 *PdMADS* genes were unevenly distributed on 8 chromosomes, with Chr 1, 2, 3, 4, 5, 6, 7, and 8 having 17, 9, 9, 3, 6, 7, 10, and 6 genes, respectively. It is worth noting that most of the *PdMADS* genes were distributed in regions with higher gene densities on these chromosomes. Similar to the findings of *MADS-box* family studies in other species, there was no significant correlation between the chromosome length and the number of *PdMADS* genes.

We further performed collinearity analysis on the *PdMADS* family members to explore the gene duplication types. A total of 16 pairs of segmental duplication genes and 9 pairs of tandem duplication genes were identified (Appendix A). A chromosomal circle diagram was drawn to show the location of the segmental genes, and the results showed that all 8 chromosomes contained segment duplication genes (Figure 4). The tandem duplication genes were distributed on four chromosomes: Chr 1, Chr 6, Chr 7 and Chr 8 (Figure 3). It is worth noting that the majority of the segment duplication genes were in the MIKC subfamily, but tandem duplication genes were mainly distributed in the Mγ subfamily. In conclusion, gene duplication events during the amplification of *PdMADS* family members may be a major mechanism.

To explore the evolutionary clues of *PdMADS* family members, we constructed a collinear map of *MADS-box* gene members in almond as well as *A. thaliana*, *O. sativa, M. domestica*, and *P. persica* (Figure 5), with 19, 3, 65, and 59 pairs of collinear genes, respectively (Appendix A). It is worth noting that multiple *PdMADS* genes had a collinear relationship with the same *MADS* gene in *M. domestica*, suggesting that these genes play an important role in the evolution of the *PdMADS* gene family. In addition, the collinear genes in almond and *P. persica* showed a one-to-one relationship with no duplicate genes, indicating that the two *MADS-box* genes had a low degree of separation evolutionarily and were highly conserved. In conclusion, gene analysis of the collinearity between almond and other species may have important implications for revealing *MADS-box* gene evolution.

We further calculated the Ka, Ks, and Ka/Ks values of segment duplication, tandem duplication, and the interspecies collinear genes of *PdMADS* family members to evaluate the selection pressures imposed on *PdMADS* family members during evolution. The segmental duplication, tandem duplication, and interspecies collinear gene Ka/Ks values among *PdMADS* members were all less than one (Appendix A). Therefore, we speculate that *PdMADS* family members may experience strong purifying selection during evolution.

### 3.5. Analysis of cis-Acting Elements in PdMADS Family Gene Promoters

A total of 112 *cis*-acting elements were coannotated from the 2000-bp upstream promoter regions of 67 *PdMADS* genes, and 66 elements had clear functional annotations (Appendix A). In addition to a large number of basic elements such as the CAAT-box and TATA-box, the promoter regions contained a variety of light-related element types, such as ACE, G-Box, and P-box. We further mapped these element types as well as the distribution of hormone regulation, stress response, and growth and developmental types (Figure 6 and Appendix A). Among them, the hormone regulatory elements included auxin (AuxRE, AuxRR-core, and TGA-box), gibberellin (TATC-box, P-box, and GARE-motif), salicylic acid (SARE), and methyl jasmonate ((TGACG)-motif). The stress response elements included five types: ARE, LTR, TC-rich repeats, MBS, and WUN-motif. There were six types of growth and development elements: CAT-box, O2-site, RY element, circadian, HD-Zip 1, and MSA-like. In conclusion, the results of the promoter region *cis*-acting elements indicate that members of the *PdMADS* family are extensively involved in the growth and development of almond.

### 3.6. Analysis of the Expression Patterns of PdMADS Family Members at Six Flowering Stages

First, we removed the *PdMADS* genes with FPKM values less than 1 at the 6 flowering stages of ‘Wanfeng’ and ‘Nonpareil’ and retained 36 genes for analysis of the expression patterns during flowering. Among them, 27 genes belonged to the MIKC subfamily, 4 to the Mα subfamily, 2 to the Mγ subfamily, and 3 to the Mδ subfamily (Appendix A). It is worth noting that the FPKM values of the *PdMADS* genes in the MIKC subfamily were generally larger, but the values for genes in the Mα, Mγ, and Mδ subgroups were smaller. According to Figure 7, *PdMADS* gene expression was upregulated in both ‘Wanfeng’ and ‘Nonpareil’ at the six flowering stages. The *PdMADS* gene expression patterns across 11 MIKC subfamily subgroups were explored. In ‘Wanfeng’, four groups of genes—FUL, SOC1, ANR1, and SVP—were mainly upregulated at the A1 stage, while four groups of genes—SEP, AGL11, APETALA3, and AGAMOUS—were mainly upregulated at stages A2, A3, A4, and A5, respectively. In ‘Nonpareil’, most of these groups of genes were upregulated toward B1 and B2 flowering. Notably, most *PdMADS* genes were downregulated in A6 and B6.

### 3.7. Fluorescence Quantitative Analysis

We selected 15 *PdMADS* genes from the MIKC subfamily to perform fluorescence quantitative analysis on 12 samples from 6 flowering stages of ‘Wanfeng’ and ‘Nonpareil’ almond cultivars (Figure 8). The results show that the expression levels of *PdMADS9*, *PdMADS35*, and *PdMADS39* were all less than or equal to 1 in the 12 samples, which suggests that they may not be involved in almond blossom stages. In addition, the expression levels of four genes—*PdMADS11*, *PdMADS12*, *PdMADS26*, and *PdMADS32*—did not change much in the 12 samples and only slightly increased in the samples at a certain flowering stage in ‘Wanfeng’ or ‘Nonpareil’. These four genes may be expressed at a certain period of time during the flowering period of almond. Three genes—*PdMADS10*, *PdMADS17*, and *PdMADS22*—had high expression levels in a certain sample at the flowering and full flowering stages, and they may be involved in the flowering process of almond. It is worth noting that five genes—*PdMADS16*, *PdMADS18*, *PdMADS23*, *PdMADS36*, and *PdMADS47*—had significant expression differences in the dormant stage of ‘Wanfeng’ and ‘Nonpareil’ almond. The expression levels in ‘Wanfeng’ and ‘Nonpareil’ almond were significantly increased at a certain time during the six flowering stages. It is speculated that these five genes play an important role in regulating the flower bud dormancy process of ‘Wanfeng’ and ‘Nonpareil’ almond and are involved in the different flowering stages of almond.

### 3.8. Protein Interaction and Enrichment Analysis of PdMADS Family Members

We predicted the potential protein interactions among *PdMADS* members using the STRING database. The results show that 25 *PdMADS* genes constituted a protein interaction network with 33 nodes, with multiple protein interactions among the nodes (Figure 9). Notably, 20 *PdMADS* genes belonged to the MIKC subfamily. Twenty-five PdMADS proteins, such as PdMADS7, PdMADS9, and PdMADS18, exhibited complex multiprotein interactions. PdMADS23 was predicted to be the central node, and 16 other genes exhibited protein interactions. In addition, GO and KEGG enrichment of the 25 interacting proteins was examined. Among them, 25 PdMADS proteins were mainly enriched in the biological process category and distributed in a variety of flower development-related terms, such as floral development (GO: 0009908), floral organ development (GO: 0048437), and floral whorl development (GO: 0048438). The KEGG results mainly show enrichment in three pathways: flower development (WP618), flower development: initiation (WP2108), and seed development (WP2279) (Appendix A).

### 3.9. MIKC Target Gene Identification and Functional Analysis

We obtained the MIKC DNA-binding sites using the JASPAR database (Appendix A). The promoter sequence 2000-bp upstream of the amygdala gene was searched, with a total of 13,515 target genes identified based on the MIKC DNA-binding sites The GO, KEGG, and domain results of the target genes were also presented (Appendix A). Among them, 10823 target genes showed GO annotation. According to Figure 10, in the biological process, the target genes were mainly enriched in GO terms such as cellular process (GO:0009987), metabolic process (GO:0008152), and single-organism process (GO:0044699). For molecular function, the target genes were mainly enriched in GO terms such as binding (GO:0005488), catalytic activity (GO:0003824), and transporter activity (GO:0005215). In the cellular component, the target genes were mainly enriched in GO terms such as cell (GO:0005623), cell part (GO:0044464), and organelle (GO:0043226) (Appendix A). Similarly, 4647 target genes were subjected to KEGG annotations, and 4204 target genes were found to be enriched in metabolism-related pathways. The top 20 KEGG pathways with the most enrichment included metabolic pathways (ko01100), biosynthesis of secondary metabolites (ko01110), and starch and sucrose metabolism (ko00500) (Figure 11 and Appendix A). We speculate that MIKC genes can affect multiple pathways by regulating downstream target genes.

## 4. Discussion

*MADS-box* genes play important roles in plant floral organ development and flowering and are involved in biotic and abiotic stress responses [3]. A variety of plant *MADS-box* gene families have been studied to date. In this study, we identified a total of 67 *MADS-box* genes in the genome of ‘Wanfeng’ almond, which is slightly lower than the amount found in plants such as *Arabidopsis* (106) [34], rice (75) [35], peach (79) [36], and apple (146) [37]. We constructed a neighbor-joining phylogenetic tree of *MADS-box* members in almond and *A. thaliana* and divided the *PdMADS* family members into four subfamilies: Mα (17), Mγ (17), Mδ (3), and MIKC (30). No *PdMADS* genes clustered in the Mβ subfamily in almond, but there were *MADS-box* genes in the Mβ subfamily in other plants, such as peach, rice, and poplar [38]. We further divided the 30 *PdMADS* genes in the MIKC subfamily into 11 groups, speculating that the regulatory mechanism of almond flowering may be the same as that of *A. thaliana*.

Gene family evolution and phylogenetic classification can be assessed based on the conserved protein domains and the structural diversity of genes [39]. We annotated 10 types of motifs in *PdMADS* family members, and the protein sequences of *PdMADS* members in the same subfamily had more conserved motif types. *PdMADS* members within the same subfamily also had a higher number of conserved exons and introns. In addition, most members of the Mα and Mγ subfamilies of type I had only one exon, and the gene structure was relatively simple. Conversely, most members of the Mδ and MIKC subgroups of type II had approximately 10 exons, and the gene structure was relatively complex. These results are consistent with the results for pineapple, wheat, and *Rhododendron Hainan* [40,41,42]. Taken together, the above results indicate that the *PdMADS* genes in types I and II are relatively conserved genetically.

Gene duplication events are the main factor leading to the rapid expansion and evolution of gene family members, and gene duplication is one of the important ways in which the number of *MADS-box* genes expands [43]. We identified 16 pairs of segmental duplications and 9 pairs of tandem duplications of *PdMADS* genes on 8 chromosomes in the ‘Wanfeng’ genome. Notably, 21 segmental duplication genes belonged to the MIKC subfamily, and 2 belonged to the Mδ subfamily. Thirteen tandem duplication genes belonged to type I. These results are consistent with those for rice and *Rhododendron Hainan*. Compared with almond, *A. thaliana*, *O. sativa, M. domestica*, and *P. persica* had 19, 3, 65, and 59 pairs of collinear *MADS-box* members, respectively. Thus, *MADS-box* genes are highly conserved based on homology during the evolution of Rosaceae. Exploring the Ka/Ks values of the duplicated genes is an efficient way to study the effects of duplicated genes on evolution. The Ka/Ks ratios of 16 pairs of segmental duplication and 9 pairs of tandem duplication genes were all less than 1, indicating that these genes underwent purifying selection during evolution. In addition, the Ka/Ks ratios of the almond, *A. thaliana, O. sativa, M. domestica*, and *P. persica MADS-box* members were less than one, indicating that the *MADS-box* genes underwent purifying selection during the evolution of different species.

The *cis*-acting elements in the promoter regions regulate gene expression. Interactions between the promoter binding sites and transcription factors play an important role in the regulation of transcriptional levels [44]. In this study, we analyzed the *cis*-acting elements in the 2000-bp upstream promoter region of *PdMADS* family members. The number of elements related to light was the largest, with each gene having approximately seven types of elements. Various elements related to growth and development, hormonal regulation, and the response to stress were detected. Examples include *cis*-acting elements and auxin-responsive elements involved in cell cycle regulation and *cis*-acting elements associated with meristem expression. We also found *cis* elements related to abiotic stress, such as those involved in defense and stress response, and others involved in low temperature response. Therefore, *cis*-acting element analysis provides important clues for the functional study of *PdMADS* family members, especially the regulation of related genes and the development of plants under different stresses.

A clear understanding of gene expression in different tissues, developmental stages, and environments is important for understanding the molecular mechanisms of biological development [45]. Therefore, we explored the expression of *PdMADS* genes across six flowering stages of ‘Wanfeng’ and ‘Nonpareil’ cultivars based on transcriptome data and observed significant differences. Especially in the MIKC subfamily, the *PdMADS* genes in the four subgroups of *SEP* (*PdMADS9* and *PdMADS25*), *FUL*, *FLC*, and *SOC1* exhibited a tendency toward upregulation at B1 and B2 in ‘Nonpareil’. *SVP* was upregulated in the sprouting stages. However, five groups—*SEP* (*PdMADS22*, *PdMADS33*, and *PdMADS42*), *AGL11*, *AGL12*, *AGAMOUS*, and *APETALA3*—were mainly upregulated in ‘Wanfeng’ in the sprouting stages. *FLC* plays an important role in the flower transition process and an inhibitory role in the autonomic and vernalization pathways of plants [46]. The expression of *PdMADS35* in the *FLC* group was three times higher in ‘Nonpareil’ than in ‘Wanfeng’. *FLC* is an inhibitor of *SOC1*. The expression level of *PdMADS39* in the *SOC1* subfamily of ‘Nonpareil’ was four times higher than that of ‘Wanfeng’. Therefore, we speculate that the later flowering of ‘Nonpareil’ in relation to ‘Wanfeng’ is mainly affected by the *FLC* and *SOC1* genes, and we will focus on exploring these genes in these two groups in a follow-up study. It is worth noting that there were fewer slightly upregulated genes in both varieties at A6 and B6, and most genes were downregulated or not expressed. Therefore, we speculate that *MADS-box* genes are not expressed at the blossom stage in almond, which should be further explored.

Plant flowering is a coherent and complex process, and flowering-related genes such as *SEP*, *APETALA*, *AG*, and *AGL* comprehensively regulate flower formation [47]. Numerous studies have shown that *FLC* can block the transcriptional activation of *SOC1* and requires *SVP* expression to delay flowering [48]. In the present study, the fluorescence quantitative results showed that the expression levels of *PdMADS47* (*SVP*) and *PdMADS16* (*AGL12*) were significantly increased in the dormant stage of ‘Nonpareil’ almond flower buds, while *PdMADS35* (*FLC*) and *PdMADS23* (*SOC1*) were expressed at a normal level in B1. It is speculated that *PdMADS47* (*SVP*) and *PdMADS16* (*AGL12*) in ‘Nonpareil’ almond may be involved in inhibiting flower buds from the vernalization process, thereby delaying flowering. In addition, 15 *PdMADS* genes had significant expression differences across the 12 samples from 6 flowering stages in ‘Wanfeng’ and ‘Nonpareil’. Their expression levels increased in a certain flowering stage during the three stages of flowering and full flowering, indicating that *PdMADS* gene expression and the regulation of the flowering process were active in almond flower buds from the dormant stages to the full flowering stages.

*PdMADS* family member protein interaction analysis can predict the functions of these genes [42]. *SOC1* is an important gene for regulating flowering, and its overexpression can suppress late flowering in plants with functional *FRI* and *FLC* alleles [48]. Notably, this study found that *SOC1* (*PdMADS23*, *26*, and *39*) interacts with multiple groups of genes, such as *SEP* (*PdMADS42*), *APETALA3* (*PdMADS61*), and *AGAMOUS* (*PdMADS36*). We speculate that the *SOC1*-type *PdMADS* gene is an important gene involved in the regulation of flowering in almond. In recent years, the research of *MADS-box* transcription factors regulating downstream target genes has made rapid progress, which can induce flower organ formation by activating downstream target genes. At present, the identified MIKC target genes include *NAC-LIKE ACTIVATED BY AP3/PI* (*NAP*), *SUPERMAN* (*SUP*), *SUPERMAN* (*SUP*), and other genes, which regulate the formation of flower organs such as petals, stamens, and ovules [49,50,51]. We identified a total of 13,515 downstream MIKC target genes in almond. The enrichment results of GO and KEGG showed that the target genes had multiple protein functions and were distributed in multiple metabolic pathways. Thus, MIKC genes affect multiple pathways through the regulation of downstream genes. The identification and preliminary exploration of these MIKC target genes in the genome of ‘Wanfeng’ almond will be important in the future.

## 5. Conclusions

In this study, we identified 67 *PdMADS* genes through the whole genome of *P. dulcis* ‘Wanfeng’ and used phylogenetic tree clustering, motif, gene structure, chromosomal location, and promoter analysis to further explore the characteristics of *PdMADS* family members. Segmental duplication is the main amplification pathway in the *PdMADS* genes family in almond, and the Ka/Ks ratios of the collinear genes indicate that these genes are mainly affected by purifying selection. Based on the transcriptome data, the expression of *PdMADS* genes in six flowering stages of ‘Wanfeng’ and ‘Nonpareil’ cultivars was explored. Finally, we explored the expression levels of 15 *PdMADS* genes in 6 flowering stages of ‘Wanfeng’ and ‘Nonpareil’ using fluorescence quantitative technology, and it was found that the difference in flowering time between ‘Wanfeng’ and ‘Nonpareil’ almond may be caused by the different expression of *MADS* genes regulating the process of vernalization. In conclusion, this study provides a reference for further research on the biological function of the *PdMADS* gene family and the mechanism of regulating flowering in flowering almond.

## Figures and Tables

**Figure 1 genes-13-01764-f001:**
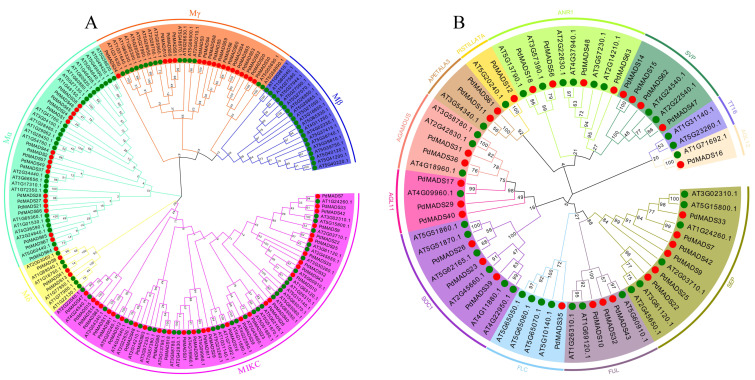
Phylogenetic tree of *MADS-box* family members in almond and *A. thaliana*. (**A**) Neighbor-joining phylogenetic tree. (**B**) Phylogenetic analysis of the MIKC subfamily *MADS-box* genes from *Arabidopsis* and almond. Each color block represents a group. Green circles represent *Arabidopsis MADS-box* genes, and red circles represent *PdMADS* genes.

**Figure 2 genes-13-01764-f002:**
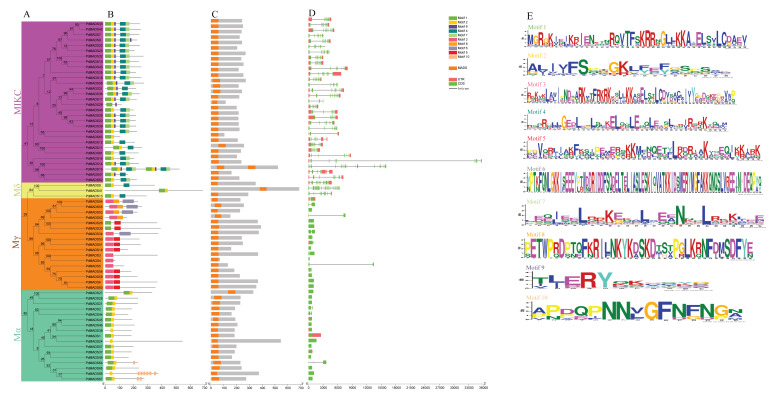
Phylogenetic clustering of *PdMADS* members based on motifs and gene structures. (**A**) Neighbor-joining phylogenetic tree of members of the *PdMADS* family. Different colored areas represent different subfamilies. (**B**) Conserved protein motifs. (**C**) MADS domains. (**D**) Gene structure: exon–intron. (**E**) Motif LOGO.

**Figure 3 genes-13-01764-f003:**
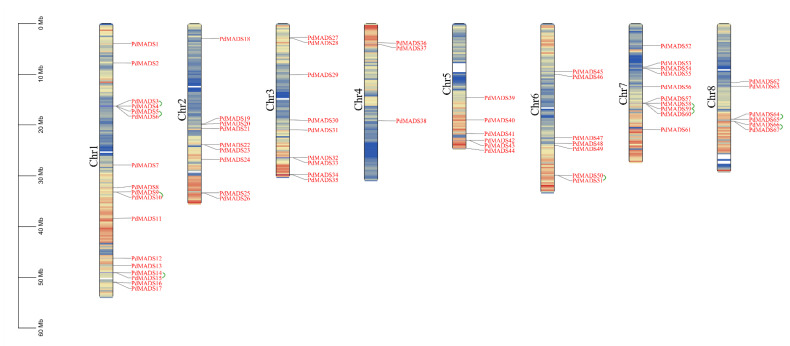
Chromosomal distribution of *PdMADS* genes in almond. The scale on the left shows the chromosome length information, and the chromosome names are listed at the top. The green lines show tandem duplication genes.

**Figure 4 genes-13-01764-f004:**
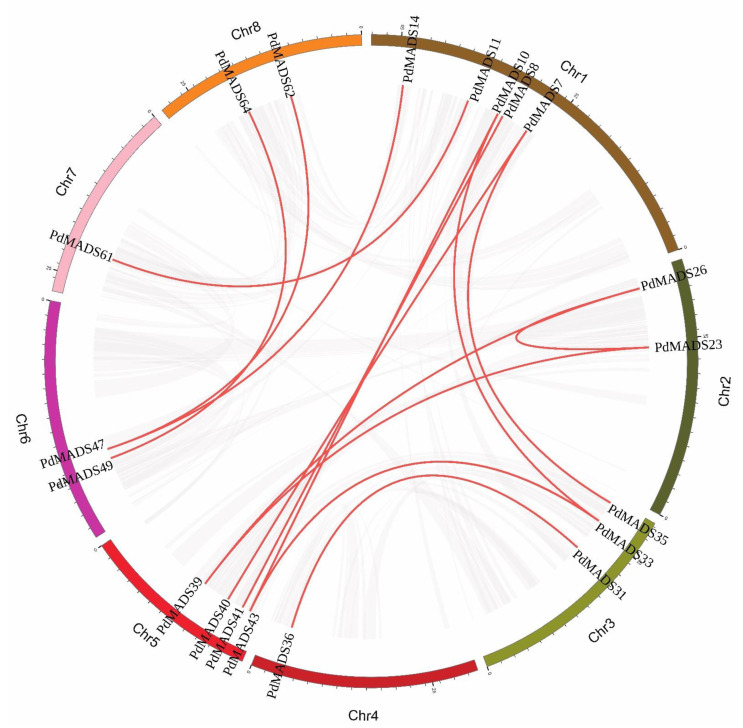
Interchromosomal relationships of *PdMADS* genes in the almond genome. The red lines link duplicated *PdMADS* gene pairs.

**Figure 5 genes-13-01764-f005:**
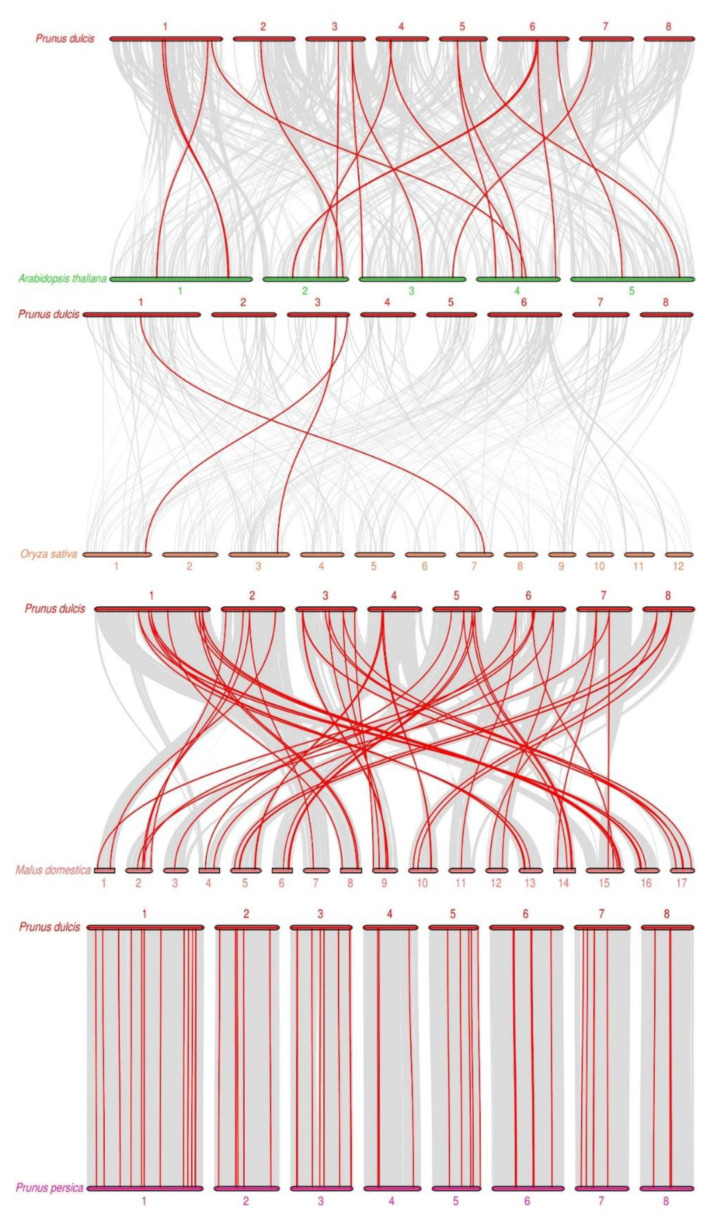
Synteny analysis of *MADS-box* family members of almond and four other species. Each horizontal line represents a chromosome, and red lines represent collinear genes.

**Figure 6 genes-13-01764-f006:**
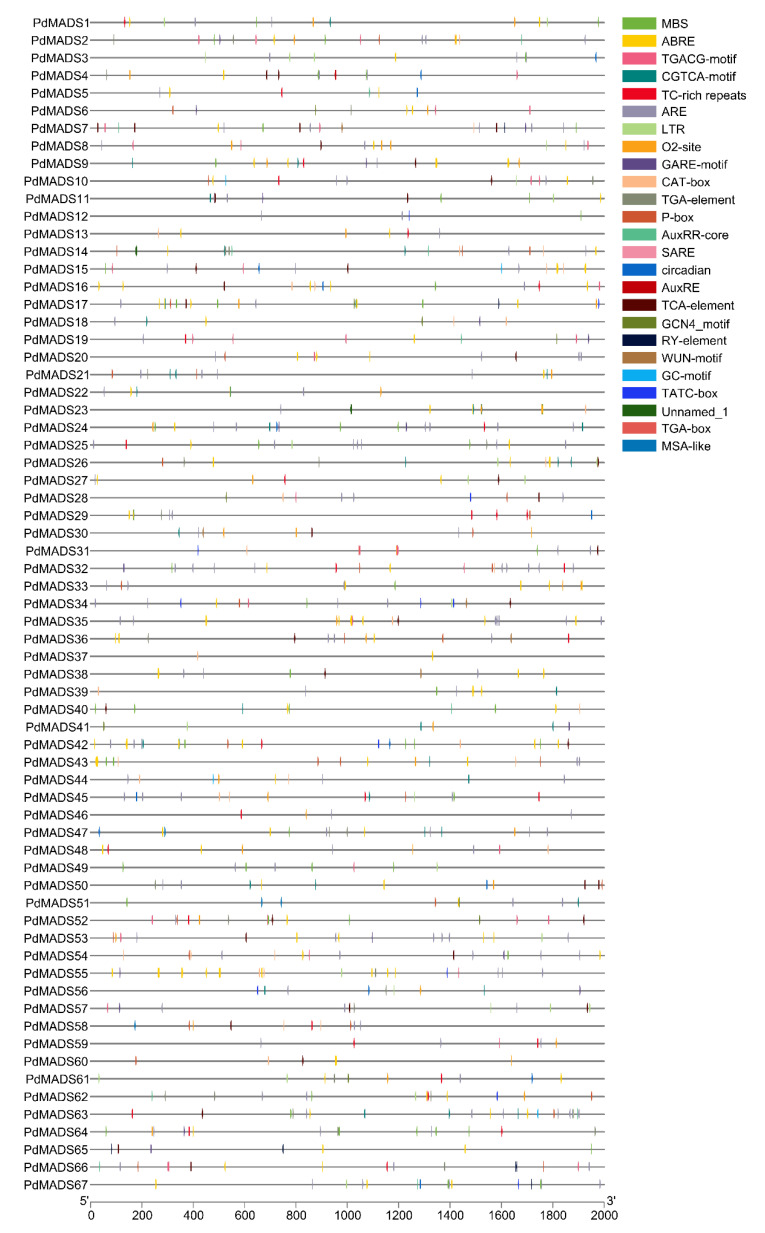
*Cis*-elements in the promoter regions of *PdMADS* members. The colored boxes indicate the types of *cis* elements.

**Figure 7 genes-13-01764-f007:**
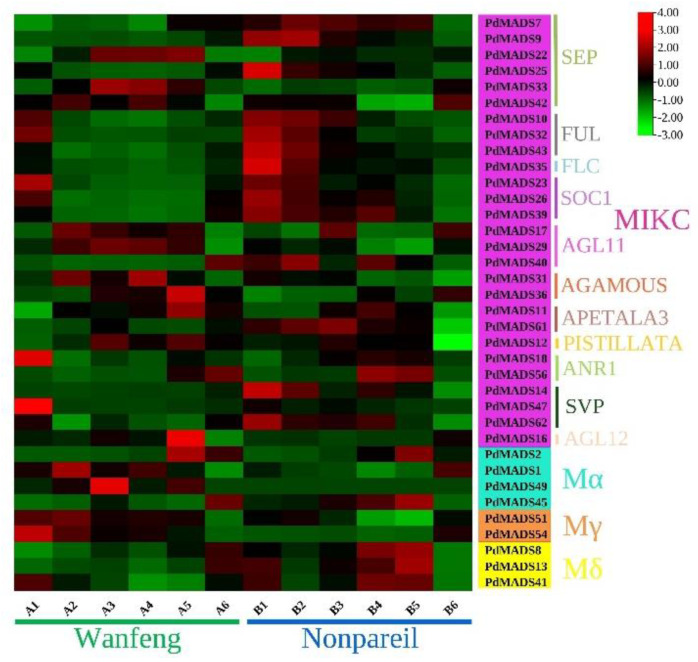
Heat map of the expression patterns of *PdMADS* family members in ‘Wanfeng’ and ‘Nonpareil’ at six flowering stages. The ROW normalization method was used to draw the heat map. Red squares indicate upregulation of expression, black squares indicate no expression, and green squares indicate downregulated expression for dormant stage (A1 and B1), sprouting stages (A2~A5 and B2~B5), and full bloom stage (A6 and B6).

**Figure 8 genes-13-01764-f008:**
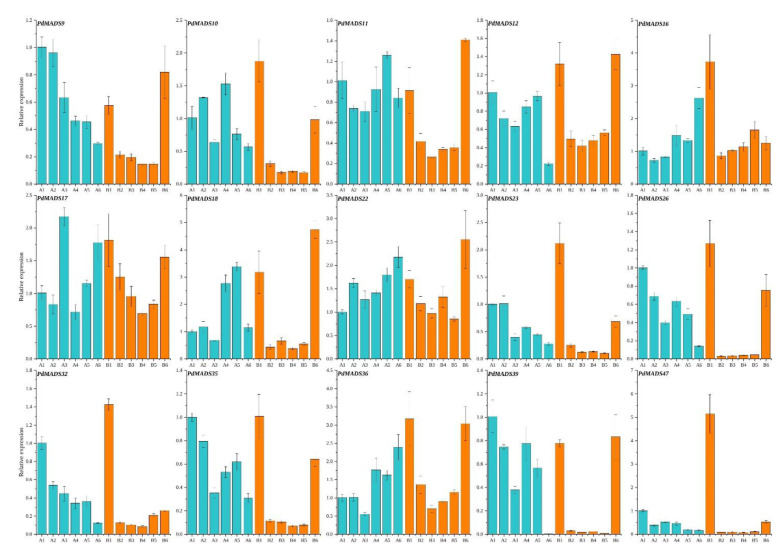
Fluorescence quantitative expression analysis of 15 *PdMADS* genes across 6 flowering stages in 2 almond cultivars: ‘Wanfeng’ and ‘Nonpareil’. Blue represents ‘Wanfeng’ almond, and yellow represents ‘Nonpareil’ almond.

**Figure 9 genes-13-01764-f009:**
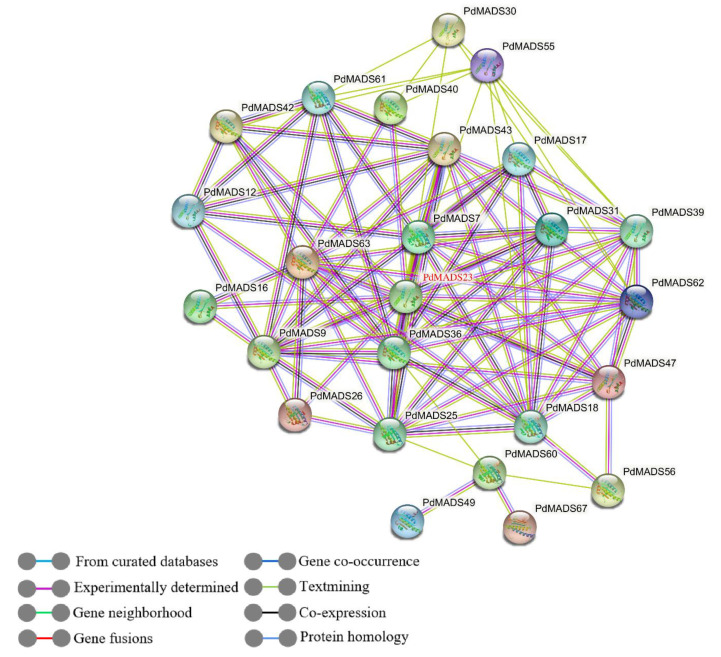
*PdMADS* member protein-protein interaction network. Gene names in red represent central genes.

**Figure 10 genes-13-01764-f010:**
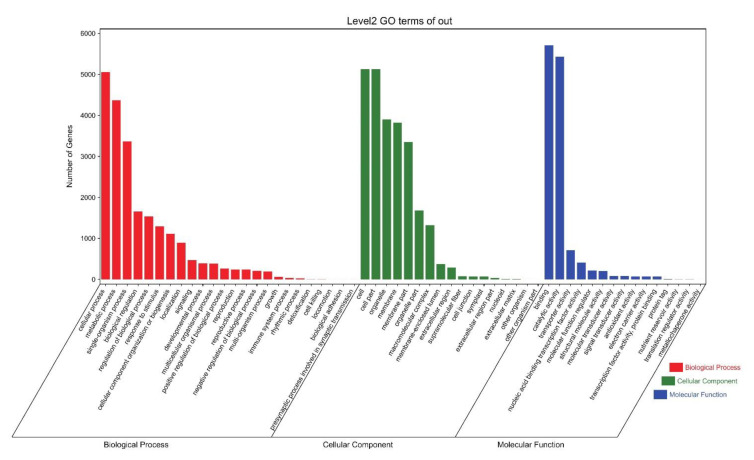
Statistics of MIKC target gene GO enrichment results.

**Figure 11 genes-13-01764-f011:**
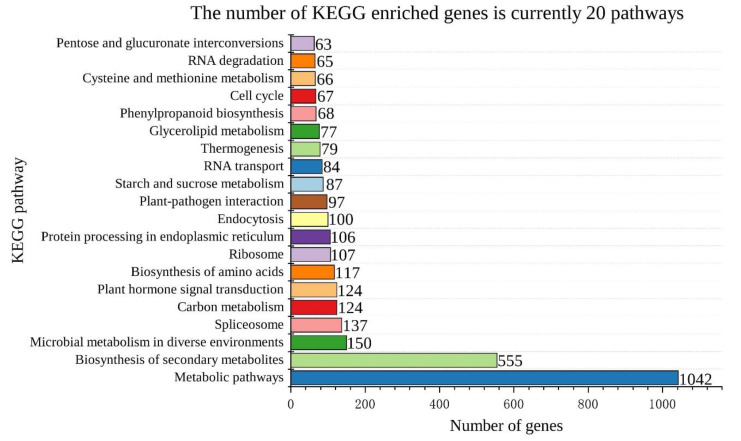
The top 20 MIKC target gene-related KEGG pathways.

## Data Availability

The data presented in this study are available in the article and its Appendix A.

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
