# Peer review of "Genome-Wide Identification and Analysis of the MADS-Box Gene Family in Almond Reveal Its Expression Features in Different Flowering Periods"

_genes, 2022, doi:10.3390/genes13101764_

Round 1

Reviewer 1 Report

Please provide more details on the methods used for RNA-seq. Currently inadequate.

Gene names for arabidopsis on the phylo trees would be helpful. And maybe assigning homologous IDs to almond genes would make it more easy for people to interpret the biology.

Reviewer 2 Report

This study Genome-wide identification and analysis of the MADS-box gene family in almond reveal its expression features in different flowering periods. Before recommending this article for publication, there are some shortcomings for that should be resolve.

How much exon and intron sequences were observed must be mention from minimum to maximum.

Also specify results in the abstract section.

Discuss economic and medicinal importance of almond in the abstract.

The importance of genome wide identification studies.

Section 2.3. should be cited with https://doi.org/10.1016/j.plaphy.2021.01.042, and section 2.7 can be cited with https://doi.org/10.1007/s10725-021-00785-7,

Results and conclusion are well presented. 
